# Dialog-based Interactive Image Retrieval

**Xiaoxiao Guo**[†]
IBM Research AI
xiaoxiao.guo@ibm.com

**Hui Wu**[†]
IBM Research AI
wuhu@us.ibm.com

**Yu Cheng**
IBM Research AI
chengyu@us.ibm.com

**Steven Rennie**
Fusemachines Inc.
srennie@gmail.com

**Gerald Tesauro**
IBM Research AI
gtesauro@us.ibm.com

**Rogerio Schmidt Feris**
IBM Research AI
rsferis@us.ibm.com

## Abstract

Existing methods for interactive image retrieval have demonstrated the merit of integrating user feedback, improving retrieval results. However, most current systems rely on restricted forms of user feedback, such as binary relevance responses, or feedback based on a fixed set of relative attributes, which limits their impact. In this paper, we introduce a new approach to interactive image search that enables users to provide feedback via natural language, allowing for more natural and effective interaction. We formulate the task of dialog-based interactive image retrieval as a reinforcement learning problem, and reward the dialog system for improving the rank of the target image during each dialog turn. To mitigate the cumbersome and costly process of collecting human-machine conversations as the dialog system learns, we train our system with a user simulator, which is itself trained to describe the differences between target and candidate images. The efficacy of our approach is demonstrated in a footwear retrieval application. Experiments on both simulated and real-world data show that 1) our proposed learning framework achieves better accuracy than other supervised and reinforcement learning baselines and 2) user feedback based on natural language rather than pre-specified attributes leads to more effective retrieval results, and a more natural and expressive communication interface.

## 1 Introduction

The volume of searchable visual media has grown tremendously in recent years, and has intensified the need for retrieval systems that can more effectively identify relevant information, with applications in domains such as e-commerce [1, 2], surveillance [3, 4], and Internet search [5, 6]. Despite significant progress made with deep learning based methods [7, 8], achieving high performance in such retrieval systems remains a challenge, due to the well-known semantic gap between feature representations and high-level semantic concepts, as well as the difficulty of fully understanding the user's search intent.

A typical approach to improve search efficacy is to allow the user a constrained set of possible interactions with the system [9, 10]. In particular, the user provides iterative feedback about retrieved objects, so that the system can refine the results, allowing the user and system to engage in a "conversation" to resolve what the user wants to retrieve. For example, as shown in Figure 1, feedback about relevance [11] allows users to indicate which images are "similar" or "dissimilar" to the desired image, and relative attribute feedback [12] allows the comparison of the desired image with candidate images based on a fixed set of attributes. While these feedback paradigms are effective, the restrictions

---

[†]These two authors contributed equally to this work.

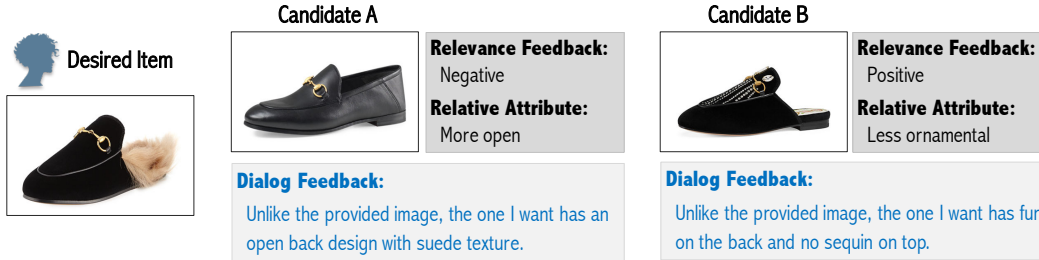

Figure 1: In the context of interactive image retrieval, the agent incorporates the user's feedback to iteratively refine retrieval results. Unlike existing work which are based on relevance feedback or relative attribute feedback, our approach allows the user to provide feedback in natural language.

on the specific form of user interaction largely constrain the information that a user can convey to benefit the retrieval process.

In this work, we propose a new approach to interactive visual content retrieval by introducing a *novel form of user feedback based on natural language*. This enables users to directly express, in natural language, the most prominent conceptual differences between the preferred search object and the already retrieved content, which permits a more natural human-computer interaction. We formulate the task as a reinforcement learning (RL) problem, where the system directly optimizes the rank of the target object, which is a non-differentiable objective.

We apply this RL based interactive retrieval framework to the task of image retrieval, which we call *dialog-based interactive image retrieval* to emphasize its capability in aggregating history information compared to existing single turn approaches [13, 14, 15, 16]. In particular, a novel end-to-end dialog manager architecture is proposed, which takes natural language responses as user input, and delivers retrieved images as output. To mitigate the cumbersome and costly process of collecting and annotating human-machine dialogs as the system learns, we utilize a model-based RL approach by training a user simulator based on a corpus of human-written relative descriptions. Specifically, to emulate a single dialog turn, where the user provides feedback regarding a candidate image relative to what the user has in mind, the user simulator generates a *relative caption* describing the differences between the candidate image and the user's desired image.[1] Whereas there is a lot of prior work in image captioning [17, 18, 19], we explore the problem of *relative image captioning*, a general approach to more expressive and natural communication of relative preferences to machines, and to use it as part of a user simulator to train a dialog system.

The efficacy of our approach is evaluated in a real-world application scenario of interactive footwear retrieval. Experimental results with both real and simulated users show that the proposed reinforcement learning framework achieves better retrieval performance than existing techniques. Particularly, we observe that feedback based on natural language is more effective than feedback based on pre-defined relative attributes by a large margin. Furthermore, the proposed RL training framework of directly optimizing the rank of the target image shows promising results and outperforms the supervised learning approach which is based on the triplet loss objective. The main contributions of this work are as follows:

- A new vision/NLP task and machine learning problem setting for interactive visual content search, where the dialog agent learns to interact with a human user over the course of several dialog turns, and the user gives feedback in natural language.

- A novel end-to-end deep dialog manager architecture, which addresses the above problem setting in the context of image retrieval. The network is trained based on an efficient policy optimization strategy, employing triplet loss and model-based policy improvement [20].

- The introduction of a computer vision task, *relative image captioning*, where the generated captions describe the salient visual differences between two images, which is distinct from and complementary to context-aware *discriminative image captioning*, where the absolute attributes of one image that discriminate it from another are described [21].

- The contribution of a new dataset, which supports further research on the task of relative image captioning. [2]

## 2 Related Work

**Interactive Image Retrieval.**    Methods for improving image search results based on user interaction have been studied for more than 20 years [22, 10, 23]. Relevance feedback is perhaps the most popular approach, with user input specified either as binary feedback ("relevant" or "irrelevant") [11] or based on multiple relevance levels [24]. More recently, relative attributes (e.g., "more formal than these," "shinier than these") have been exploited as a richer form of feedback for interactive image retrieval [12, 25, 26, 27, 28]. All these methods rely on a fixed, pre-defined set of attributes, whereas our method relies on feedback based on natural language, providing more flexible and more precise descriptions of the items to be searched. Further, our approach offers an end-to-end training mechanism which facilitates deployment of the system in other domains, without requiring the explicit effort of building a new vocabulary of attributes.

**Image Retrieval with Natural Language Queries.**    Significant progress has been recently made on methods that lie in the intersection of computer vision and natural language processing, such as image captioning [18, 19], visual question-answering [29, 30], visual-semantic embeddings [31, 32], and grounding phrases in image regions [33, 34]. In particular, our work is related to image or video retrieval methods based on natural language queries [13, 14, 15, 16]. These methods, however, retrieve images and videos in a single turn, whereas our proposed approach aggregates history information from dialog-based feedback and iteratively provides more refined results.

**Visual Dialog.**    Building conversational agents that can hold meaningful dialogs with humans has been a long-standing goal of Artificial Intelligence. Early systems were generally designed based on rule-based and slot-filling techniques [35], whereas modern approaches have focused on end-to-end training, leveraging the success of encoder-decoder architectures and sequence-to-sequence learning [36, 37, 38]. Our work falls into the class of visually-grounded dialog systems [39, 40, 41, 42, 43]. Das et al [39] proposed the task of visual dialog, where the system has to answer questions about images based on a previous dialog history. De Vries et al. [40] introduced the *GuessWhat* game, where a series of questions are asked to pinpoint a specific object in an image, with restricted answers consisting of yes/no/NA. The *image guessing* game [42] demonstrated emergence of grounded language and communication among visual dialog agents with no human supervision, using RL to train the agents in a goal-driven dialog setting. However, these dialogs are purely text-based for both the questioner and answerer agents, whereas we address the interactive image retrieval problem, with an agent presenting images to the user to seek feedback in natural language.

## 3 Method

Our framework , which we refer to as the *dialog manager*, considers a user interacting with a retrieval agent via iterative dialog turns. At the $t$-th dialog turn, the dialog manager presents a candidate image $a_t$ selected from a retrieval database $\mathcal{I} = \{I_i\}_{i=0}^N$ to the user. The user then provides a feedback sentence $o_t$, describing the differences between the candidate image $a_t$ and the desired image. Based on the user feedback and the dialog history up to turn $t$, $H_t = \{a_1, o_1, ..., a_t, o_t\}$, the dialog manager selects another candidate image $a_{t+1}$ from the database and presents it to the user. This process continues until the desired image is selected or the maximum number of dialog turns is reached. In practice, the dialog manager could provide multiple images per turn to achieve better retrieval performance. In this work, we focus on a simplified scenario with a single image per interaction. We note that the same framework could be extended to the multiple-image case by allowing the user to select one image out of a list of candidate images to provide feedback on.

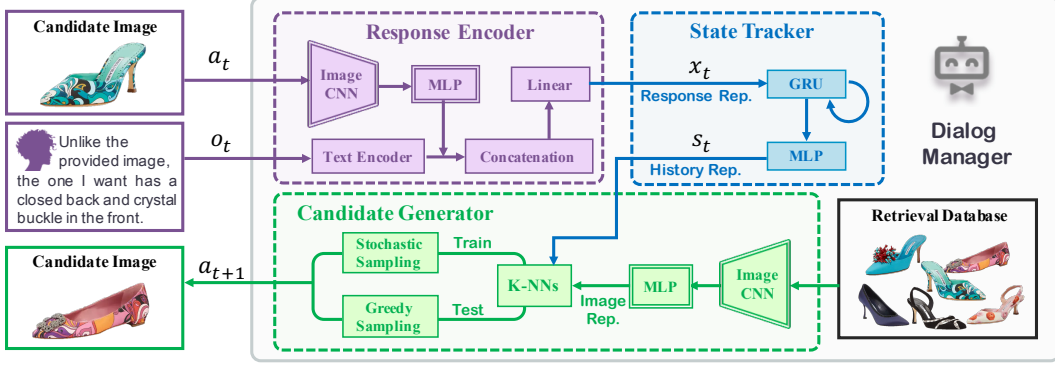

Figure 2: The proposed end-to-end framework for dialog-based interactive image retrieval.

## 3.1 Dialog Manager: Model Architecture

Our proposed dialog manager model consists of three main components: a *Response Encoder*, a *State Tracker*, and a *Candidate Generator*, as shown in Figure 2. At the $t$-th dialog turn, the *Response Encoder* embeds a candidate image and the corresponding user feedback $\{a_t, o_t\}$ into a joint visual-semantic representation $x_t \in \mathbb{R}^D$. The *State Tracker* then aggregates this representation with the dialog history from previous turns, producing a new feature vector $s_t \in \mathbb{R}^D$. The *Candidate Generator* uses the aggregated representation $s_t$ to select a new candidate image $a_{t+1}$ that is shown to the user. Below we provide details on the specific design of each of the three model components.

**Response Encoder.** The goal of the Response Encoder is to embed the information from the $t$-th dialog turn $\{a_t, o_t\}$ into a visual-semantic representation $x_t \in \mathbb{R}^D$. First, the candidate image is encoded using a deep convolutional neural network (CNN) followed by a linear transformation: $x_t^{\text{im}} = \text{ImgEnc}(a_t) \in \mathbb{R}^D$. The CNN architecture in our implementation is an ImageNet pre-trained ResNet-101 [44] and its parameters are fixed. Words in the user feedback sentence are represented with one-hot vectors and then embedded with a linear projection followed by a CNN as in [45]: $x_t^{\text{txt}} = \text{TxtEnc}(o_t) \in \mathbb{R}^D$. Finally, the image feature vector and the sentence representation are concatenated and embedded through a linear transformation to obtain the final response representation at time $t$: $x_t = W(x_t^{\text{im}} \oplus x_t^{\text{txt}})$, where $\oplus$ is the concatenation operator and $W \in \mathbb{R}^{D \times 2D}$ is the linear projection. The learnable parameters of the Response Encoder are denoted as $\theta_r$.

**State Tracker.** The State Tracker is based on a gated recurrent unit (GRU), which receives as input the response representation $x_t$, combines it with the history representation of previous dialog turns, and outputs the aggregated feature vector $s_t$. The forward dynamics of the State Tracker is written as: $g_t, h_t = \text{GRU}(x_t, h_{t-1})$, $s_t = W^s g_t$, where $h_{t-1} \in \mathbb{R}^D$ and $g_t \in \mathbb{R}^D$ are the hidden state and the output of the GRU respectively, $h_t$ is the updated hidden state, $W^s \in \mathbb{R}^{D \times D}$ is a linear projection and $s_t \in \mathbb{R}^D$ is the history representation updated with the information from the current dialog turn. The learnable parameters of the State Tracker (GRU model) are denoted as $\theta_s$. This memory-based design of the State Tracker allows our model to sequentially aggregate the information from user feedback to localize the candidate image to be retrieved.

**Candidate Generator.** Given the feature representation of all images from the retrieval database, $\{x_i^{\text{im}}\}_{i=0}^N$, where $x_i^{\text{im}} = \text{ImgEnc}(I_i)$, we can compute a sampling probability based on the distances between the history representation $s_t$ to each image feature, $x_i^{\text{im}}$. Specifically, the sampling probability is modeled using a softmax distribution over the top-$K$ nearest neighbors of $s_t$: $\pi(j) = e^{-d_j} / \sum_{k=1}^K e^{-d_k}, j = 1, 2, ..., K$, where $d_k$ is the L2 distance of $s_t$ to its $k$-th nearest neighbor in $\{x_i^{\text{im}}\}_{i=0}^N$. Given the sampling distribution, two approaches can be taken to sample the candidate image, denoted as $a_{t+1} = I_{j'}$: (1) stochastic approach (used at training time), where $j' \sim \pi$, and (2) greedy approach (used at inference time), where $j' = \arg\max_j(\pi_j)$. Combining the three components of the model architecture, the overall learnable parameters of the dialog manager model is $\theta = \{\theta_r, \theta_s\}$. Next, we explain how the network is trained end-to-end.

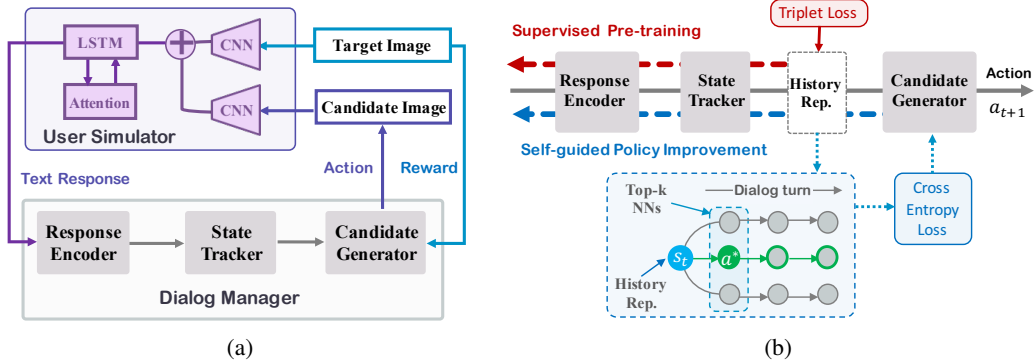

Figure 3: The learning framework: (a) The user simulator enables efficient exploration of the retrieval dialogs (Section 3.2.1); and (b) the policy network is learned using a combination of supervised pre-training and model-based policy improvement (Section 3.2.2).

## 3.2 Training the Dialog Manager

Directly optimizing the ranking percentile metric in a supervised learning scheme is challenging since it is a non-differentiable function [46, 47]. Instead, we model the ranking percentile as the environment reward received by the agent and frame the learning process in a reinforcement learning setting with the goal of maximizing the expected sum of discounted rewards: $\max_{\pi} \mathcal{U}^{\pi} = \mathbb{E}\left[\sum_{t=1}^{T} \gamma^{t-1} r_t | \pi_\theta\right]$, where $r_t \in \mathbb{R}$ is the reward representing the ranking percentile of the target image at the $t$-th interaction, $\gamma$ is a discount factor determining the trade-off between short-term and long-term rewards, $T$ is the maximum number of dialog turns, and $\pi_\theta$ is the policy determined by network parameters $\theta$.[3]

Training an RL model for this problem requires extensive exploration of the action space, which is only feasible if a large amount of training data is available. However, collecting and annotating human-machine dialog data for our task is expensive. This problem is exacerbated in the situation of natural language based user feedback, which incurs an even larger exploration space as compared to approaches based on a fixed set of attributes. In text-based dialog systems, it is common to circumvent this issue by relying on user simulators [48]. We adopt a similar strategy, where a user simulator, trained on human-written relative descriptions, substitutes the role of a real user in training the dialog manager (illustrated in Figure 3a). Below we further describe our user simulator, as well as the reinforcement learning techniques that we used to optimize our dialog manager.

### 3.2.1 User Simulator based on Relative Captioning

Here we propose the use of a *relative captioner* to simulate the user. It acts as a surrogate for real human users by automatically generating sentences that can describe the prominent visual differences between any pair of target and candidate images. We note that at each turn, our user simulator generates feedback independent of previous user feedback, and previously retrieved images. While more sophisticated models that consider the dialog history could potentially be beneficial, training such systems well may require orders of magnitude more annotated data. In addition, back-referencing in dialogs can inherently be ambiguous and complex to resolve, even for humans. Based on these considerations, we decided to first investigate the use of a single-turn simulator. While a few related tasks have been studied previously, such as context-aware image captioning [21] and referring expression generation [49], to the best of our knowledge, there is no existing dataset directly supporting this task, so we introduce a new dataset as described in Section 4.

We experimented with several different ways of combining the visual features of the target and retrieved images. We include a comprehensive study of different models for the user simulator in Appendix C and show that the relative captioner based user model serves as a reasonable proxy for real users. Specifically, we used feature concatenation to fuse the image features of the target and the reference image pair and applied the model of *Show, Attend, and Tell* [50] to generate the relative captions using a long short-term memory network (LSTM). For image feature extraction, we adopted

the architecture of ResNet101 [44] pre-trained on ImageNet; and to better capture the localized visual differences, we added a visual attention mechanism; the loss function of the relative captioner is the sum of the negative log likelihood of the correct words [50].

### 3.2.2 Policy Learning

**Supervised Pre-training.** When the network parameters are randomly initialized at the beginning, the history representations $s_t$ are nearly random. To facilitate efficient exploration during RL training, we first pre-train the policy using a supervised learning objective. While maximum likelihood-based pre-training is more common, here we pre-train using the more discriminative triplet loss objective:

$$\mathcal{L}^{\text{sup}} = \mathbb{E}\Big[ \sum_{t=1}^{T} \max(0, \|s_t - x^+\|_2 - \|s_t - x^-\|_2 + \text{m})\Big] \tag{1}$$

where $x^+$ and $x^-$ are the image features of the target image and a random image sampled from the retrieval database respectively, m is a constant for the margin and $\|.\|_2$ denotes $L2$-norm. Intuitively, by ensuring the proximity of the target image and the images returned by the system, the rank of the target image can be improved without costly policy search from random initialization. However, entirely relying on this supervised learning objective deviates from our main learning objective, since the triplet loss objective does not jointly optimize the set of candidate images to maximize expected future reward. [4]

**Model-Based Policy Improvement.** Given the known dynamics of the environment (in our case, the user simulator), it is often advantageous to leverage its behavior for policy improvement. Here we adapt the policy improvement [20] to our model-based policy learning. Given the current policy $\pi$ and the user simulator, the value of taking an action $a_t$ using test-time configuration can be efficiently computed by look-ahead policy value estimation $Q^\pi(h_t, a_t) = \mathbb{E}\big[ \sum_{t'=t}^{T} \gamma^{t'-t} r_{t'} | \pi \big]$. Because our user simulator is essentially deterministic, one trajectory is sufficient to estimate one action value. Therefore, an improved policy $\pi'$ can be derived from the current policy $\pi$ by selecting the best action given the value of the current policy, $\pi'(h_t) \equiv a_t^* = \arg\max_a Q^\pi(h_t, a)$. Specifically, following [51], we minimize the cross entropy loss given the derived action, $a^*$,

$$\mathcal{L}^{\text{imp}} = \mathbb{E}\Big[ -\sum_{t=1}^{T} \log \Big( \pi(a_t^* | h_t) \Big) \Big] \tag{2}$$

Compared to traditional policy gradient methods, the model-based policy improvement gradients have lower variance, and converge faster. In Section 5, we further demonstrated the effectiveness of model-based policy improvement by comparing it with a recent policy gradient method. Figure 3b illustrates our policy learning method as described above.

## 4    Dataset: Relative Captioning

Our user simulator aims to capture the rich and flexible language describing visual differences of any given image pair. The relative captioning dataset thus needs this property. We situated the data collection procedure in a scenario of a shopping chatting session between a shopping assistant and a customer. The annotator was asked to take the role of the customer and provide a natural expression to inform the shopping assistant about the desired product item. To promote more regular, specific, and relative user feedback, we provided a sentence prefix for the annotator to complete when composing their response to a retrieved image. Otherwise the annotator response is completely free-form: no other constraints on the response were imposed. We used Amazon Mechanical Turk to crowdsource the relative expressions. After manually removing erroneous annotations, we collected in total $10,751$ captions, with one caption per pair of images.

Interestingly, we observed that when the target image and the reference image are sufficiently different, users often directly describe the visual appearance of the target image, rather than using relative expressions (c.f. fourth example in Figure 7(b), Appendix A). This behavior mirrors the *discriminative captioning* problem considered in [21], where a method must take in two images

and produce a caption that refers only to one of them. Relative and discriminative captioning are complementary, and in practice, both strategies are used, and so we augmented our dataset by pairing 3600 captions that were discriminative with additional dissimilar images. Our captioner and dialog-based interactive retriever are thus trained on both discriminative and relative captions, so as to be respectively more representative of and responsive to real users. Additional details about the dataset collection procedure and the analysis on dataset statistics are included in Appendix A and Appendix B.

## 5  Experimental Results

In Section 5.1, we assess the contribution of each component of our pipeline for policy learning. To evaluate the value of using free-form dialog feedback, we show experiments considering both simulated user feedback (Section 5.2) and real-world user feedback (Section 5.3).

All experiments were performed on the *Shoes* dataset [53], with the same training and testing data split for all retrieval methods and for training the user simulator. $10,000$ database images were used during training, and $4,658$ images for testing. The retrieval models are tested by retrieving images on the testing set, starting from a randomly selected candidate image for the first dialog turn. Image retrieval performance is quantified by the average rank percentile of the image returned by the dialog manager on the test set. For details on architectural configurations, parameter settings, baseline implementation, please refer to Appendix D.

### 5.1  Analysis of the Learning Framework

We use our proposed user simulator to generate data and provide extensive quantitative analysis on the contribution of each model component.

**Results on Relative Captioner.**  Figure 5 provides examples of simulator generated feedback and the collected user annotations. An interesting observation is that even though the user simulator only occasionally generates descriptions that exactly match the human annotations (the third example in Figure 5), it can still summarize the main visual differences between the images, since inherently there are many ways to describe differences between two images. Qualitative examination of the generated relative expressions showed that the user simulator can approximate feedback of real users at a very low annotation cost (more analysis is included in Appendix C).

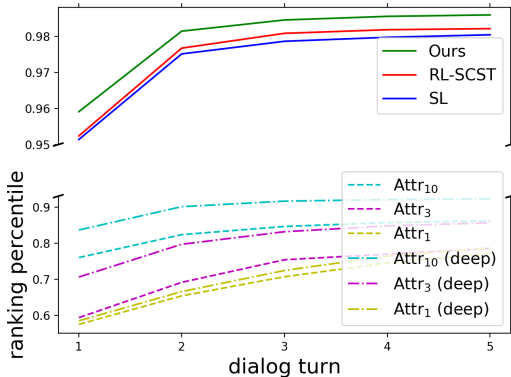

Figure 4: Quantitative comparison of our method and two baselines and methods using feedback based on a pre-defined set of relative attributes.

**Policy Learning Results.**  To investigate how retrieval performance is affected by each component of the dialog manager, we compare our approach, denoted as **Ours**, against two variants: (1) **SL**: supervised learning where the agent is trained only with triplet loss; (2) **RL-SCST**: policy learning using Self-Critical Sequence Training (SCST) [19] after pre-training the network using the triplet loss objective. As shown in Figure 4 (solid lines), the average ranking percentile of the target image in all methods increases monotonically as the number of dialog turns increases. Both RL-based retrieval algorithms outperform the supervised pre-training, **SL**, which is expected since the triplet loss function does not directly optimize the retrieval ranking objective. Finally, **Ours** achieves $98\%$ average ranking percentile with only two dialog turns and consistently outperforms **RL-SCST** across different dialog turns, which demonstrates the benefit of the model-based policy improvement component. We have observed similar results on the attribute-based baselines. Each of the SL based attribute model underperforms its RL version by $\sim 1\%$ in retrieval ranking percentile.

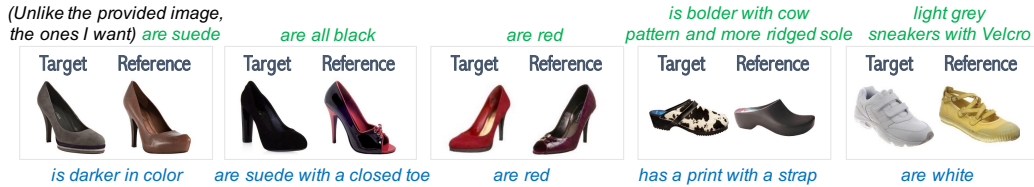

Figure 5: Examples of human provided (green) and captioner generated relative descriptions (blue). While generated relative captions don't resemble human annotations in most cases, they can nonetheless capture the main visual differences between the target image and the reference image.

## 5.2 Effectiveness of Natural Language Feedback

In this section, we empirically evaluate the effect of natural language feedback, compared to pre-defined, relative attribute-based user feedback.

**Generating Attribute Feedback.** Each image in the dataset maps to a 10-D attribute vector, as described in [12]. We adopted a rule-based feedback generator which concatenates the respective attribute words with "more" or "less", depending on the relative attribute values of a given image pair. For example, if the "shiny" value of the candidate image and the target image are $0.9$ and $1.0$ respectively, then the rule-based feedback is "more shiny." Attributes are randomly sampled, similar to the relative attribute feedback generation in [12]. To simulate the scenario when users provide feedbacks using multiple attributes, individual attribute phrases are combined. We adopted original attribute values in [12], which were predicted using hand-crafted image features, as well as attribute values predicted using deep neural networks in [54].

**Results.** We trained the dialog manager using both dialog-based feedback and attribute-based feedback ($\textbf{Attr}_n$ and $\textbf{Attr}_n$ (**deep**)), where the subscript number denotes the number of attributes used in the rule-based feedback generator and (**deep**) denote baselines using deep learning based attribute estimates as in [54]. The empirical result is summarized in Figure 4, including relative attribute feedback using one, three and ten attribute phrases. The three attribute case matches the average length of user feedback in free-form texts and the ten case uses all possible pre-defined attributes to provide feedback. Across different numbers of dialog turns, the natural language based agent produced significantly higher target image average ranking percentile than the attribute based methods. The results suggest that feedback based on unrestricted natural language is more effective for retrieval than the predefined set of relative attributes used in [12]. This is expected as the vocabulary of relative attributes in [12] is limited. Even though deep learning based attribute estimates improve the attribute-based baselines significantly, the performance gap between attribute based baseline and free form texts is still significant. We conjecture that the main reason underlying the performance gap between attribute and free-form text based models is the effectively open domain for attribute use, which is difficult to realize in a practical user interface without natural language. In fact, free-form dialog feedback obviates constructing a reliable and comprehensive attribute taxonomy, which in itself is a non-trivial task [55].

## 5.3 User Study of Dialog-based Image Retrieval

In this section, we demonstrate the practical use of our system with real users. We compare with an existing method, WhittleSearch [12], on the task of interactive footwear retrieval. WhittleSearch represents images as feature vectors in a pre-defined 10-D attribute space, and iteratively refines retrieval by incorporating the user's relative feedback on attributes to narrow down the search space of the target image. For each method, we collected 50 five-turn dialogs; at each turn, one image is presented to the user to seek feedback. For WhittleSearch, the user can choose to use any amount of attributes to provide relative feedback on during each interaction. The resulting average ranking percentile of the dialog manager and WhittleSearch are $89.9\%$ and $70.3\%$ respectively. In addition to improved retrieval accuracy, users also reported that providing dialog-based feedback is more natural compared to selecting the most relevant attributes from a pre-defined list.

Figure 6 shows examples of retrieval dialogs from real users (please refer to Appendix E for more results and discussions). We note that users often start the dialog with a coarse description of the

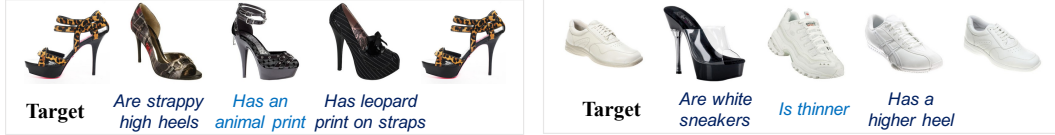

Figure 6: Examples of the user interacting with the proposed dialog-based image retrieval framework.

main visual features (color, category) of the target. As the dialog progresses, users give more specific feedback on fine-grained visual differences. The benefit of free-form dialog can be seen from the flexible usage of rich attribute words ("leopard print on straps"), as well as relative phrases ("thinner", "higher heel"). Overall, these results show that the proposed framework for the dialog manager exhibits promising behavior on generalizing to real-world applications.

## 6   Conclusions

This paper introduced a novel and practical task residing at the intersection of computer vision and language understanding: dialog-based interactive image retrieval. Ultimately, techniques that are successful on such tasks will form the basis for the high fidelity, multi-modal, intelligent conversational systems of the future, and thus represent important milestones in this quest. We demonstrated the value of the proposed learning architecture on the application of interactive fashion footwear retrieval. Our approach, enabling users to provide natural language feedback, significantly outperforms traditional methods relying on a pre-defined vocabulary of relative attributes, while offering more natural communication. As future work, we plan to leverage side information, such as textual descriptions associated with images of product items, and to develop user models that are conditioned on dialog histories, enabling more realistic interactions. We are also optimistic that our approach for image retrieval can be extended to other media types such as audio, video, and e-books, given the performance of deep learning on tasks such as speech recognition, machine translation, and activity recognition.

**Acknowledgement.** We would like to give special thanks to Professor Kristen Grauman for helpful discussions.

## Footnotes

[1]In this work, the user simulator is trained on single-turn data and does not consider the dialog history. This reduces the sequence of responses to a "bag" of responses and implies that all sequences of a given set of actions (candidate images) are equivalent. Nevertheless, while the set of candidate images that maximize future reward (target image rank) are a set, selecting the image for the next turn naturally hinges on all previous feedback from the user. Therefore, the entire set of candidate images can be efficiently constructed sequentially.

[2]The project website is at: `www.spacewu.com/posts/fashion-retrieval/`

[3]Strictly speaking, the optimal policy depends on the number of remaining dialog turns. We simplify the policy to be a function independent of dialog turn numbers.

[4]More explanation on the difference between the two objectives is provided in Appendix E.

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
