[Supplementary Material]

# Supplemental Material: Dialog-based Interactive Image Retrieval

## A   Data Collection

In the following, we explain the details on how we collected the relative captioning dataset for training the user simulator and provide insights on the dataset properties. Unlike existing datasets which aim to capture the visual differences purely using "more" or "less" relations on visual attributes [12], we want to collect data which captures comparative visual differences that are hard to describe merely using a pre-defined set of attributes. As shown in Figure 8, we designed the data collection interface in the context of fashion footwear retrieval, where a conversational shopping assistant interacts with a customer and whose goal is to efficiently retrieve and present the product that matches the user's mental image of the desired item.

Figure 7: Length distribution of the relative captioning dataset (a), and examples of relative captions collected in the dataset (b). The leading phrase "*Unlike the provided image, the ones I want*" is omitted for brevity.

**Collecting Relative Expressions.** The desired annotation for relative captioning should be free-form and introduce minimum constraints on how a user might construct the feedback sentence. On the other hand, we want the collected feedback to be concise and relevant for retrieval and avoid casual and non-informative phrases (such as "*thank you*", "*oh, well*"). Bearing the two goals in mind, we designed a data collection interface as shown in Figure 8, which provided the beginning phrase of the user's response ("*Unlike the provided ...*") and the annotators only needed to complete the sentence by giving an informative relative expression. This way, we can achieve a balance between sufficient lexical flexibility and avoiding irrelevant and casual phrases. After manual data cleaning, we are left with $10,751$ relative expressions with one annotation per image pair.

**Augmenting Dataset with Single-Image Captions.** During our data collection procedure for relative expressions, we observed that when the target image and the reference image are visually distinct (fourth example in Figure 7(b)), users often only implicitly use the reference image by directly describing the visual appearance of the target image. Inspired by this, we asked annotators to give direct descriptions on 3600 images without the use of reference images. We then paired each image in this set with multiple visually distinct reference images (selected using deep feature similarity). This data augmentation procedure further boosted the size of our dataset at a relatively low annotation cost.

## B   Dataset analysis

Figure 7(a) shows the length distribution of the collected captions. Most captions are very concise (between 4 to 8 words), yet composing a large body of highly rich vocabularies as shown in Figure 9 [5] . Interestingly, although annotators have the freedom to give feedback in terms of comparison on a single visual attribute (such as "*is darker*", "*is more formal*"), most feedback expressions consist of compositions of multiple phrases that often include spatial or structural details (Table 1).

Figure 8: AMT annotation interface. Annotators need to assume the role of the customer and complete the rest of the response message. The collected captions are concise, and only contain phrases that are useful for image retrieval.

Figure 9: Visualization of the rich vocabulary discovered from the relative captioning dataset. The size of each rectangle is proportional to the word count of the corresponding word.

| Single Phrase (36%) | Composition of Phrases (63%) | Propositional Phrases (40%) |
|---|---|---|
| are brownish | is more athletic and is white | is lower on the ankle and blue |
| have a zebra print | has a larger sole and is not a high top | have rhinestones across the toe and a strap |
| have a thick foot sheath | has lower heel and exposes more foot and toe | are brown with a side cut out |
| are low-top canvas sneakers | is white, and has high heels, not platforms | is in neutrals with buckled strap and flatter toe |
| have polka dot linings | is alligator, not snake print, and a pointy tip | is more rugged with textured sole |

Table 1: Examples of relative expressions. Around two thirds of the collected expressions contain composite feedback on more than one types of visual feature. And 40% of the expressions contain propositional phrases that provide information containing spatial or structural details.

Figure 10: Ratings of relative captions provided by humans and different relative captioner models. The raters were asked to give a score from 1 to 4 on the quality of the captions: no errors (4), minor errors (3), somewhat related (2) and unrelated (1).

Examples of the collected relative expressions are shown in Figure 7(b). We observed that, in some cases, users apply a concise phrase to describe the key visual difference (first example); but most often, users adopt more complicated phrases (second and third examples). The benefit of using free-form feedback can be seen in the second example: when the two shoes are exactly the same on most attributes (white color, flat heeled, clog shoes), the user resorts to using composition of a fine-grained visual attribute ("*holes*") with spatial reference ("*on the top*"). Without free-form dialog based feedback, this intricate visual difference would be hard to convey.

## C Human Evaluation of Relative Captioning Results

We tested a variety of relative captioning models based on different choices of feature fusion and the use of attention mechanism. Specifically, we tested one *Show and Tell* [18] based model, **RC-FC** (using concatenated deep features as input), and three *Show, Attend and Tell* [50] based models, including **RC-FCA** (feature concatenation), **RC-LNA** (feature fusion using a linear layer) and **RC-CNA** (feature fusion using a convolutional layer). For all methods, we adopted the architecture of ResNet101 [44] pre-trained on ImageNet to extract deep feature representation.

We report several common quantitative metrics to compare the quality of generated captions in Table 2. Given the intrinsic flexibility in describing visual differences between two images, and the lack of comprehensive variations of human annotations for each pair of images, we found that common image captioning metrics does not provide reliable evaluation of the actual quality of the generated captions.

| Target | Reference | | Target | Reference | | Target | Reference | | Target | Reference | | Target | Reference |

| Unlike the provided image, the one(s) I want **are brown with a pointy toe** | are brown leather with a top buckle | are red , with a lower heel | are burgundy , not black | are black patent leather |

| Unlike the provided image, the one(s) I want **are blue and green sneakers** | *are floral print with an all-over floral pattern* | are brown with a higher heel | are black with a thicker heel | *are purple and black* |

Figure 11: Examples of generated relative captions using **RC-FCA**. Red fonts highlight inaccurate or redundant descriptions.

Table 2: Quantitative metrics of generated relative captions on Shoes dataset.

|        | BLEU-1 | BLEU-4 | ROUGE |
|--------|--------|--------|-------|
| RC-CNA | 32.5   | 11.2   | 45.4  |
| RC-LNA | 30.7   | 10.7   | 43.2  |
| RC-FCA | 29.6   | 10.3   | 42.9  |
| RC-FC  | 26.3   | 8.8    | 40.4  |

Therefore, to better evaluate the caption quality, we directly conducted human evaluation, following the same rating scheme used in [18]. We collected user ratings on relative captions generated by each model and those provided by humans on 1000 image pairs. Both quantitative results and human evaluation (Figure 10) suggest that all relative captioning models produced similar performance with **RC-CNA** exhibiting marginally better performance. It is also noticeable that there is a gap between human provided descriptions and all automatically generated captions, and we observed some captions with incorrect attribute descriptions or are not entirely sensible to humans, as shown in Figure 11. This indicates the inherent complexity of task of relative image captioning and room for improvement of the user simulator, which will lead to more robust and generalizable dialog agents.

## D   Experimental Configurations

Since no official training and testing data split was reported on *Shoes* dataset, we randomly selected $10,000$ images as the training set, and the rest $4,658$ images as the held-out testing set. The user simulator adopts the same training and testing data split as our dialog manager: it was trained using image pairs sampled from the training set with no overlap with the testing images. Since the four models for relative image captioning produced similar qualitative results in the user study, we selected **RC-FCA** model as our user simulator since it leads to more efficient training time for the dialog manager than the **RC-CNA** model. The baseline method, **RL-SCST**, uses the same network architecture and the same supervised pre-training step as our dialog manager and also utilizes the user simulator for training. The idea of **RL-SCST** is to use test-time inference reward as the baseline for policy gradient learning by encouraging policies performing above the baseline while suppressing policies under-performing the baseline. Given the trained user simulator, we can easily compute the test-time rewards for **RL-SCST** by greedy decoding rather than stochastically sampling the image to return at each dialog turn.

For all methods, the embedding dimensionality of the feature space is set to $D = 256$; the MLP layer of the image encoder is finetuned using the single image captions to better capture the domain-specific image features. For **SL** training, we used the ADAM optimizer with an initial learning rate of $0.001$ and the margin parameter $m$ is set to $0.1$. For all reinforcement learning based methods, we employed the RMSprop optimizer with an initial learning rate of $10^{-5}$, and the discount factor is set to 1. For our dialog manager, we set the number of nearest neighbors as 3 for the Candidate Generator.

Figure 12: Examples of users interacting with the proposed dialog manager system. User feedbacks are shown below the corresponding images. "*Unlike the provided image, the ones I want*" is omitted from each sentence for brevity.

# E   Discussions on the Dialog Manager

In this section, we provide more discussions on the proposed dialog manager framework and point out a few directions for improvement.

**Dialog-based User Interaction.**

Figure 12 shows more examples of the dialog interactions on human users. In all examples, the target image reached a final ranking within the top 100 images (about 97% in ranking percentile) within five dialog turns. These examples indicate that, visible improvement of retrieval results often comes from a flexible combination of direct reference to distinctive visual attributes of the target image, and comparison to the candidate image based on relative attributes. Ideally, feedback based on a pre-defined attribute set can achieve similar performance if the attribute vocabulary is sufficiently comprehensive and descriptive (which often consists of hundreds of words as in our footwear retrieval application). But in practice, it is infeasible to ask the user to scroll through a list of hundreds of attribute words and select the optimal one to provide feedback on.

Figure 13: Illustration of the triple loss objective and the ranking objective.

Further, we observe that the system tends to be less responsive to certain low-frequency words generated by the use simulator (such as "slouchy" in the third example). This is as expected, since

the dialog manager is trained on the user simulator, which in itself has limitations (such as the fixed size of vocabulary after being trained, and the lack of memory for dialog history). We are interested in finetuning the dialog manager on real users, so that it can directly adapt to new vocabularies from the user. In summary, results on real users demonstrated that free-form dialog feedback is able to capture various types of visual differences with great lexical flexibility and can potentially result in valuable applications in real-world image retrieval systems.

**Dialog Manager Learning Framework.** One main advantage of the proposed RL based framework is to train the agent end-to-end with a non-differentiable objective function (the target image rank). While triplet loss based objective makes it efficient to pre-train the dialog manager, it still deviates from the ranking objective. As illustrated in Figure 13: two examples exhibit similar triplet loss objectives, but the target image ranks differ greatly.

We noticed that the dialog manager based on the current learning architecture sometimes forgets information from past turns. For example, in the second example of Figure 12, the second turn imposes a "yellow accents" requirement to the target image. While this feedback is reflected in the immediate next turn, it is missing from the later turns of the dialog. We think that model architectures which better incorporates the dialog history is able to alleviate this issue. We could in principle investigate more variations of the network design to further improve its performance. Overall, the proposed network architecture is effective in demonstrating the applicability of dialog-based interactive image retrieval.

## Footnotes

[5]A few high-frequency words are removed from this chart, including "has/have", "is/are", "a", "with".