[Reviews · NeurIPS 2018]

Reviewer 1



Update: Thank you for your feedback. Given your comments as well as the discussion with the other reviewers I have slightly adjusted my score to be more positive. I still stand by my comments below in that I think the work is interesting, but that the presentation in its current form is misleading. Assuming the paper will be accepted, I implore you to reconsider the presentation of this work, particularly with respect to claiming that this is a fully fledged dialogue system. The idea that the last image chosen represents a distillation of all previous rounds of dialogue is fanciful. Likewise, as pointed out by several reviews, this work is strongly related to a prior art on the subject of relative and incremental captioning. I would appreciate an effort to better represent the state of those fields in your introduction and background sections. Original review: This paper discusses a dialog based image retrieval system as well as a user simulator that this system is trained with. The dialog here is multi-modal in that while the user speaks, the retrieval system only returns a single image in lieu of an answer. The model for this problem looks reasonable, combining encodings for the previous candidate image and the user input, feeding these into a recurrent state model and subsequently proposing candidates using a KNN lookup. The user simulator is somewhat less inspired, ignoring dialog history and simply generating a sentence given a target and a previous candidate image. In the evaluation of the paper, the proposed model is shown to perform well compared with an ablation study of simpler models. All of this looks fine, and there are some nice bits about performance gains through model-based RL in the discussion section, but in balance there is not enough in this paper to merit a higher score in my opinion. The task is very specific without a good justification of why we should care about it or convincing pointers on how the research proposed here would be relevant to other problems. Likewise, the dataset is quite small, and lacking more detailed analysis of train/test results; the quality of the user simulator model is difficult to judge (which seems to spit out surprisingly good answers on presumably examples from a validation set, considering the limited size of training data), and the remaining results are difficult to put into perspective considering the only comparison is with the ablation study and some strongly crippled baselines. Unfortunately with the task and analysis falling short, the model itself does not provide enough novelty to really motivate accepting this paper in this format. I feel the current paper would be an OK workshop or short paper, but falls short of what would be expected of a long paper.

Reviewer 2



Post Rebuttal and Discussion Update ============================ I remain in support of this paper and think the community would find it interesting. That said, I agree with the other reviewers that more careful, qualified use of the word dialog would benefit the paper and leave scope for future work which does allow references to past rounds. ============================ This work presents an interactive dialog-driven image retrieval problem and explores it in the context of shoe retrieval. While logically only a small step away from caption or phrase based retrieval, the interactive nature of the problem leads to interesting challenges. This problem is well motivated and will likely be of interest to the community at large. The paper is well written and provides substantial detail regarding design decisions of both the proposed method and the data collection process (additional details in supplement). Overall the results point towards the proposed method working quite well, as do the supplied video demos. 1] A minor gripe is with respect to novelty of 'relative image captioning' defined as the task of 'generat[ing] captions that describe the salient visual differences between two images'. The work "Context-aware Captions from Context-agnostic Supervision" [51] presents a very similar problem called 'discriminative image captioning'. In this task, methods must take in two images and produce a caption which refers only to one of them. As the authors note in the appendix, a common user mode from the dataset is to simply describe the target image independent of the retrieved image when they are sufficiently different. This portion of the problem seems to mirror discriminative captioning more than relative captioning. For other comments that describe target visual features relative to the retrieved image, relative captioning is a distinct problem. 2] It would have been nice to see a SL based attribute model for completeness. Minor: L164 - "Training an RL model for this problem requires extensive exploration of the action space, which is only feasible if a large amount of training data is available." I do not understand this point, why does RL training require substantial training data collection? Is this in reference to SL pretraining cost? Or user-in-the-loop training? I do find myself wondering how well these models and training regimes would extend beyond such a narrow scope (product images of shoes) but this is surely outside the scope of this submission.

Reviewer 3



This paper proposes a novel approach to interactive visual content retrieval. It formulates the task as a reinforcement learning problem, and rewards the dialog system for improving the rank of the target image during each dialog turn. To alleviate the cost of collecting human-machine conversations as the dialog system learns, the dialog system is trained with a user simulator, which is itself trained to describe the differences between target and retrieved images. Experiments show the advantage of the proposed RL-based dialog approach over alternative baselines. [strengths] 1. To my knowledge, this is the first dialog-based interactive image retrieval system where the system and user communicate using natural language. The proposed framework and its components are appropriate for the problem. 2. The idea of using a user simulator for relative image captioning is interesting. This helps avoid the costly alternative which would be to collect human-machine dialogs as the system learns. 3. A new dataset for the task of relative image captioning is introduced. However, the usefulness of the dataset may be limited (see below, weaknesses point 2). [weaknesses] 1. The paper's conclusion that "user feedback based on natural language rather than pre-specified attributes leads to more effective retrieval results" is too strong given the weakness of the pre-specified attributes baseline (Sec. 5.2). To me, this is the biggest weakness of the paper. Specifically, if my understanding is correct, that baseline uses a 10-D attribute vector, which is computed using hand-crafted features like GIST and color histograms [13]. Thus, it is not clear whether the improvement is due to natural language feedback (as the paper claims) or if it's due to better features. This aspect needs to be studied in more detail. 2. For the relative image captioning dataset to be truly useful, a more systematic and automatic way of evaluation should be presented that goes beyond user studies (as presented in Appendix C). This is important so that new methods that try to tackle this problem can be properly compared. 3. It'd be helpful if the paper could provide some typical failure cases so that the reader can have a better understanding of the system. [summary] Overall, due to the novelty of the problem and sound approach, my initial rating is to accept the paper. However, there are also some weaknesses (as mentioned above) that I would like to see addressed in the rebuttal. [post-rebuttal final comments] Overall, the rebuttal has addressed to some degree the weaknesses stated in my initial review. I am, however, not completely satisfied with the authors' response regarding my point that the paper's claim "user feedback based on natural language rather than pre-specified attributes leads to more effective retrieval results" is too strong. Any comparison with hand-crafted feature based approaches needs to be stated clearly as so, as it is difficult to draw other conclusions out of such a comparison; i.e., if the baseline were to use a deep representation, it may perform just as well as the proposed approach. This needs to be empirically verified. Although this weakness is not enough for me to lower my initial rating, I feel that the paper either needs to (1) perform the appropriate experiment (i.e. the baseline should use a 10-D attribute vector derived from a deep neural network) or (2) clearly state that the baselines' features are hand-crafted and thus much weaker and may be the key cause for its lower performance. The rebuttal states that the revised paper will have something along the lines of (2), but I think it's important to explicitly state that the baseline is using "hand-crafted features".